# Construction of High-Density Genetic Map and QTL Mapping for Grain Shape in the Rice RIL Population

**DOI:** 10.3390/plants12162911

**Published:** 2023-08-10

**Authors:** Minyi Wei, Tongping Luo, Dahui Huang, Zengfeng Ma, Chi Liu, Yuanyuan Qin, Zishuai Wu, Xiaolong Zhou, Yingping Lu, Liuhui Yan, Gang Qin, Yuexiong Zhang

**Affiliations:** 1Guangxi Key Laboratory of Rice Genetics and Breeding, Rice Research Institute, Guangxi Academy of Agricultural Sciences, Nanning 530007, China; m18275845028@163.com (M.W.); ttp168128@sina.com (T.L.); hdh1103@163.com (D.H.); 13737942181@163.com (Z.M.); liuchi66@126.com (C.L.); 5930008@163.com (Z.W.); tlldv7735283@163.com (X.Z.); yanliuhui1995@163.com (L.Y.); 2State Key Laboratory for Conservation and Utilization of Subtropical Agro-Bioresources, Nanning 530004, China; 3Agricultural Science and Technology Information Research Institute, Guangxi Academy of Agricultural Sciences, Nanning 530007, China; qyy3931@163.com; 4Liuzhou Branch, Guangxi Academy of Agricultural Sciences, Liuzhou Research Center of Agricultural Sciences, Liuzhou 545000, China; m17671620451@163.com

**Keywords:** rice, recombinant inbred lines, high-density genetic map, grain shape, QTL mapping

## Abstract

Grain shape is an important agronomic trait directly associated with yield in rice. In order to explore new genes related to rice grain shape, a high-density genetic map containing 2193 Bin markers (526957 SNP) was constructed by whole-genome resequencing of 208 recombinant inbred (RILs) derived from a cross between ZP37 and R8605, with a total genetic distance of 1542.27 cM. The average genetic distance between markers was 0.76 cM, and the physical distance was 201.29 kb. Quantitative trait locus (QTL) mapping was performed for six agronomic traits related to rice grain length, grain width, length-to-width ratio, thousand-grain weight, grain cross-sectional area, and grain perimeter under three different environments. A total of 39 QTLs were identified, with mapping intervals ranging from 8.1 kb to 1781.6 kb and an average physical distance of 517.5 kb. Among them, 15 QTLs were repeatedly detected in multiple environments. Analysis of the genetic effects of the identified QTLs revealed 14 stable genetic loci, including three loci that overlapped with previously reported gene positions, and the remaining 11 loci were newly identified loci associated with two or more environments or traits. Locus 1, Locus 3, Locus 10, and Locus 14 were novel loci exhibiting pleiotropic effects on at least three traits and were detected in multiple environments. Locus 14, with a contribution rate greater than 10%, influenced grain width, length-to-width ratio, and grain cross-sectional area. Furthermore, pyramiding effects analysis of three stable genetic loci showed that increasing the number of QTL could effectively improve the phenotypic value of grain shape. Collectively, our findings provided a theoretical basis and genetic resources for the cloning, functional analysis, and molecular breeding of genes related to rice grain shape.

## 1. Introduction

Rice is one of the world’s most important staple crops, providing sustenance for over half of the global population. However, the current level of rice production is still insufficient to meet the demands of population growth. It is projected that the overall demand for rice will increase in the coming decades, particularly in Africa and Asia [1,2,3]. Therefore, enhancing rice yield continues to be a major focus in the field of rice breeding [4,5]. Grain shape is an important agronomic trait closely associated with yield characteristics in rice [6]. Accurate genetic analysis of grain shape traits holds significant theoretical value for high-yield and high-quality molecular breeding in rice.

Grain shape traits in rice primarily include grain length (GL), grain width (GW), grain length-to-width ratio (GLWR), and so on. These traits are typically quantitative in nature, influenced by multiple genes, although a few are controlled by single or double genes [7,8]. Studies have shown that genes controlling different grain shape traits exhibit complementary and cumulative effects [9,10]. To date, over 100 genes/quantitative trait loci (QTLs) related to rice grain shape traits have been reported, distributed across almost all rice chromosomes. Examples of genes predominantly controlling GL include *GS3* [11], *GL3.1* [12], *GL4* [13], and *GLW7* [14]. Genes mainly controlling GW include *GW2* [15], *GW5* [16], *GW8* [17], and *GS5* [18]. Additionally, there are other important genes, such as *TGW6* [19], *TGW2* [20], and *qTGW3* [21], which primarily affect rice yield by regulating grain shape and thousand-grain weight (TGW).

Some QTLs that control grain traits exhibit pleiotropy, simultaneously affecting multiple traits related to rice grains. For example, the *GL7*/*GW7* gene located on chromosome 7 influences both GL and GW. Upregulation of *GL7* leads to decreased transverse cell division and increased longitudinal cell division, resulting in thinner and longer grains with improved appearance [22,23]. Another example is the *GL2*/*GS2* gene located on chromosome 2, which is a major QTL that simultaneously influences GL, GW, and grain weight in rice [9,24]. Additionally, genes such as *GW6a* [25] and *GS9* [26] also exhibit pleiotropic effects and play crucial roles in grain shape. The localization, cloning, and functional analysis of genes related to rice grain shape have significant implications for improving yield and enhancing grain quality.

With the rapid development of high-throughput sequencing technology in recent years, the cost of sequencing has continuously decreased. Single nucleotide polymorphism (SNP) has emerged as a third-generation molecular marker technology, which allows for the rapid acquisition of a large number of polymorphic markers in a short period of time. It has become a preferred choice for constructing high-density genetic maps due to its advantages of high throughput, high marker density, and time and labor efficiency. This technology provides an effective means for exploring and identifying important agronomic trait QTLs, facilitating subsequent fine mapping and breeding applications [27].

In recent years, a number of researchers have localized rice grain shape QTLs by using a high-density genetic map constructed by whole genome resequencing of recombinant inbred lines in rice [28,29,30]. In addition, an increasing number of researchers have used high-density genetic linkage maps to identify and discover some other important agronomic trait genes/QTLs. For example, Chen et al. [31] have used the GBS strategy to detect 85,743 high-quality SNP markers, constructing a high-density genetic map containing 2711 recombinant bin markers. The average physical distance between markers is 137.68 kb, and a total of 12 significant QTL clusters affecting grain shape and endosperm chalkiness are detected. Among them, four QTLs are consistent with previously reported positions, while eight are new QTL clusters. In 2021, Yang et al. [32] utilized this high-density genetic map to identify 16 additive loci associated with early seedling vigor (ESV), three of which are stable QTLs. Jin et al. [33] have constructed a high-density genetic map with a total length of 2456.4 cM, containing 3830 SNP markers, with an average genetic distance of 0.82 cM between markers. Fifteen QTLs related to rice grain quality with LOD scores ≥ 4 are identified. Yang et al. [34] have used GBS technology to construct a high-density genetic map with 2498 bin markers, with an average physical distance of only 149.38 kb between markers. They have detected a total of 20 QTLs for anaerobic tolerance at the germination and bud stages, with six loci overlapping with those in previous reports and nine loci being novel.

In this study, we aimed to construct a high-density bin genetic map by performing whole-genome resequencing of two superior parents, R8605 and ZP37, along with their hybrid-derived population consisting of 208 RILs. Subsequently, we evaluated the grain-related traits of the RIL population in three distinct environments. Leveraging the high-density genetic map, we successfully identified novel and consistent QTLs associated with rice grain-related traits. The ultimate goal of this research was to establish a foundation for the cloning, functional analysis, and molecular breeding of genes involved in rice grain-related traits.

## 2. Results

### 2.1. Phenotypic Variation in the ZP37 X R8605 RIL Population

This study examined and statistically analyzed six grain-related traits of the parents ZP37, R8605, and the 208 RILs in three different environments (Table 1). The results showed that compared to ZP37, R8605 exhibited increased values for all grain-related traits. Analysis of variance indicated significant or extremely significant differences between the two parents for GL, GLWR, TGW, PL, and AS. However, the difference in GW was only significant in the 2019 environment and not in the other two environments. Overall, the significant differences in grain-related traits between ZP37 and R8605 indicated substantial genetic variation between the parents, which was favorable for QTL identification. Frequency distribution and the results of skewness and kurtosis tests for each grain-related trait showed wide variation in the ZP37/R8605 RIL population. The coefficient of variation for the six traits ranged from 4.00% to 11.08% across the three environments. Among them, the coefficient of variation for TGW was relatively higher, ranging from 10.17% to 11.08%, indicating that TGW was more susceptible to environmental influences. All six traits exhibited approximately normal or near-normal continuous distributions in the RIL population, indicating that they were quantitative traits controlled by multiple genes. The trends of variation were similar across the three environments, and there was evidence of transgressive segregation in the RIL population, which was consistent with the requirements for QTL mapping.

### 2.2. Correlation Analysis

Figure 1 displays the pairwise phenotypic correlations between the six grain-related traits in the three different environments. There were some variations in the correlation coefficients among different ecological environments, while the differences were minor. The results across the three environments were highly similar, indicating significant or extremely significant correlations between the grain-related traits. GL and GW analysis revealed no significant correlation for the simple traits, suggesting that GL and GW had different genetic bases. GL exhibited extremely significant positive correlations with GLWR, TGW, PL, and AS, with an exceptionally high correlation with PL (r ≥ 0.98). GW showed significant or extremely significant correlations with GLWR, TGW, PL, and AS, with the highest correlation observed with AS (r ≥ 0.83), while the correlation with PL was relatively low (r ≤ 0.18). For the composite trait GLWR, all five traits had varying degrees of impact, with GW having the most considerable influence (r ≤ −0.77). Regarding TGW, both GW and AS exhibited extremely significant positive correlations with large correlation coefficients (r ≥ 0.68). GL also showed an extremely significant correlation with TGW, indicating that GW was the most influential simple trait of TGW in this population. At the same time, GL also had some impact on TGW.

### 2.3. Sequencing Data Analysis and Construction of the High-Density Genetic Map

Using the GBS method, we conducted whole-genome resequencing on ZP37/R8605 parents and their derived 208 RIL progenies. ZP37 and R8605 parents yielded 12.75 Gbp and 12.21 Gbp of clean data, respectively. The total data volume for the 208 RILs was 972.76 Gbp, with an average of 4.68 Gbp per family. The Q30 score reached over 80% for all samples. A total of 804,031 polymorphic SNPs were detected between the parents, which were filtered and refined to obtain 526,957 high-quality, biallelic, and homozygous SNPs. These SNP markers were converted into 2193 bin markers, which were assigned to 12 linkage groups corresponding to the 12 chromosomes of rice. The total genetic map distance of the 12 chromosomes was 1542.27 cM, with marker numbers ranging from 71 to 308, averaging 182.75 per chromosome. The genetic distances ranged from 68.68 to 193.02 cM, with an average genetic distance of 128.52 cM. The average genetic and physical distances between adjacent bin markers were 0.76 cM and 201.29 kb, respectively (Figure 2A,B and Table 2). Among them, chromosomes 3 and 4 had the highest number of bin markers, with 293 and 308 markers, respectively. Chromosomes 6 and 10 had fewer bin markers, with 88 and 71 markers, respectively. Chromosomes 1 and 10 had the most considerable average map distance, measuring 1.03 and 1.10 cM, respectively, while chromosome 3 had the smallest average map distance of 0.47 cM. Table 2 presents the essential information regarding marker numbers, total map distance, and average map distance for each chromosome. The Spearman correlation coefficients between marker genetic and physical map distances were close to 1 (0.99965), indicating significant collinearity between genetic markers and the genome (Figure 2C). The constructed high-density genetic map demonstrated high quality and met the requirements for QTL mapping.

### 2.4. QTL Mapping for Grain Shape Traits

QTL analysis was performed using the constructed high-density genetic map and phenotypic data of six grain-related traits in 208 RIL progenies and their parents across three environments. QTLs with overlapping confidence intervals for the same trait in different environments were grouped together as the same QTL. A total of 39 QTLs associated with grain morphology were identified (Table 3), distributed across chromosomes 2, 3, 4, 5, 7, and 8. The mapping intervals ranged from 8.1 kb to 1781.6 kb, with an average physical distance of 517.5 kb. The LOD scores ranged from 2.85 to 12.30, and the percentage of variance explained (PVE) of individual QTLs ranged from 1.45% to 14.20%. Among them, 15 QTLs were repeatedly detected in more than two different environments. Both positive and negative additive effects were observed in the identified QTLs, indicating that both parents contributed favorable alleles.

Most of the detected QTLs were located within the same interval or in close proximity. The specific results for each grain-related trait QTL localization were as follows:

GL: In the three environments, five major-effect QTLs were identified on chromosomes 2, 7, and 8. The PVE ranged from 1.74% to 6.96%. Among them, two QTLs, *qGL-7-1*, and *qGL-7-3* were detected in a single environment, while *qGL-2*, *qGL-7-2*, and *qGL-8* were stable QTLs detected in multiple environments. The other QTLs showed positive additive effects except for *qGL-2*, which exhibited a negative additive effect. This finding indicated that the beneficial allele for *qGL-2* originated from R8605, while the beneficial alleles for the other five QTLs were derived from ZP37.

GW: A total of five GW QTLs were detected across chromosomes 3, 4, and 8 in the three environments. The PVE of individual QTLs ranged from 3.23% to 14.10%. Among them, two QTLs, *qGW-3-1* and *qGW-4-1*, were detected in a single environment, while *qGW-3-2*, *qGW-4-2*, and *qGW-8* were stable QTLs detected in multiple environments. The two QTLs on chromosome 4 exhibited negative additive effects, while the three QTLs on chromosomes 3 and 8 showed positive additive effects. It is worth noting that *qGW-8* on chromosome 8 was not only detected in all three environments but also had a consistently high PVE (PVE ≥ 9.21), and its beneficial allele originated from ZP37.

GLWR: A total of six GLWR QTLs were detected across chromosomes 2, 3, 4, 7, and 8 in the three environments. The PVE of individual QTLs ranged from 1.45% to 14.20%. Among them, five QTLs (*qGLWR-2-1*, *qGLWR-2-2*, *qGLWR-4*, *qGLWR-5*, and *qGLWR-7*) were detected in a single environment, while the QTL *qGLWR-8* was a stable QTL repeatedly detected in all three environments, with contribution percentages of 14.20%, 13.83%, and 12.46%, respectively. It was a high-contributing QTL, and its beneficial allele originated from ZP37.

TGW: A total of eight QTLs for TGW were detected across chromosomes 2, 4, 5, 7, and 8 in the three environments. The PVE of individual QTLs ranged from 2.09% to 5.75%. Seven QTLs (*qTGW-2*, *qTGW-4-1*, *qTGW-4-2*, *qTGW-4-3*, *qTGW-7-1*, *qTGW-7-2*, and *qTGW-8-2*) were detected in a single environment, while two QTLs (*qTGW-3* and *qTGW-8-1*) were stable QTLs detected in multiple environments. Except for *qTGW-2* and *qTGW-3*, which exhibited positive additive effects, the other seven QTLs showed negative additive effects. This finding indicated that the beneficial alleles for increasing TGW mainly originated from R8605.

AS: A total of nine QTLs were detected on chromosomes 02, 03, 04, 07, and 08 across three different environments. The PVE of individual QTLs ranged from 2.45% to 8.08%. Six of these QTLs, namely *qAS-2-1*, *qAS-2-2*, *qAS-4-2*, *qAS-4-3*, *qAS-4-4*, and *qAS-7*, were detected in a single environment, while three QTLs, *qAS-3*, *qAS-4-1*, and *qAS-8*, were identified as stable QTLs across multiple environments. In addition, five of these QTLs associated with the AS trait displayed negative additive effects, while four showed positive additive effects, indicating that both parental lines contributed favorable alleles.

PL: Only three QTLs for PL were detected across three different environments on chromosomes 02, 07, and 08. The PVE of individual QTLs ranged from 3.12% to 6.91%. All three QTLs were identified as stable QTLs across multiple environments. Among them, *qPL-7* and *qPL-8* exhibited negative additive effects, while *qPL-2* displayed a positive additive effect, indicating that both parental lines contributed favorable alleles.

### 2.5. Analysis of Loci with Stable Genetic Effects of Grain Shape

Through genetic effect analysis of all identified 39 grain-related QTLs, a total of 14 loci with stable genetic effects were obtained, involving 30 QTLs (Table 4). Among them, three loci overlapped with previously reported genes/QTLs, while the remaining 11 loci were newly discovered. Locus 2 was identified in the analysis of grain cross-sectional area traits in 2020. It was located within the interval of Block1352-Block13543 on chromosome 2, with a contribution rate of 5.34%. The corresponding physical position was 31,122,113-31,447,924 bp. A positive regulator, *PGL2* [35], is associated with rice GL and GW within this interval. Locus 8 was mapped as a QTL for GLWR in the 2020 environment. It was located within the interval of Block4850-Block4851 on chromosome 5, with a contribution rate of 3.53%. The physical position was 27,634,637-27,858,389 bp. Within this interval, there is a gene *OsPUP7* [36] that simultaneously controls rice plant height, number of grains per panicle, and grain shape cytokinin transporter. Locus 13 (*qGL-8*, *qTGW-8-2*, and *qPL-8*) was a pleiotropic locus that affected GL, TGW, and PL. It was mapped to the interval of Block5989-Block6012 on chromosome 8, with a physical distance of 22,065,235-23,447,755 bp. Within this position, a G protein gamma subunit *GGC2* [37] regulates rice GL and GW.

The other 11 newly identified loci in this study were detected in different environments, indicating their stable genetic effects and highlighting the value and reliability of high-density genetic maps for QTL analysis. Locus 4 and Locus 12 were repeatedly detected as QTLs for a single trait in two environments. The remaining nine loci were associated with multiple traits, among which Locus 1, Locus 3, Locus 10, and Locus 14 exhibited genetic pleiotropy, controlling three or more traits and detected across multiple environments. Locus 1, located within the interval of Block1305–Block1318 on chromosome 2, corresponded to the physical position of 29,677,627–30,291,539 bp. It simultaneously influenced traits such as GL, TGW, and PL. Locus 3, located within the interval of Block1968–Block1974 on chromosome 3, corresponded to the physical position of 16,148,731–16,617,377 bp. It simultaneously affected traits such as GW, TGW, and AS. Locus 10, located within the interval of Block5433–Block5490 on chromosome 7, corresponded to the physical position of 20,606,977–22,680,384 bp. It simultaneously influenced traits such as GL, TWG, AS, and PL.

Notably, Locus 14 had the most prominent genetic effect, with a contribution rate exceeding 10%. It was located within the interval of Block6167–Block6169 on chromosome 8, corresponding to the physical position of 27,555,403–27,911,017 bp. It simultaneously affected traits such as GW, GLWR, and AS.

### 2.6. Analysis of Genetic Pleiotropy and Pyramiding Effects of Three Grain-Related Loci

This study selected three QTLs with stable genetic effects, namely Locus 1, Locus 10, and Locus 13, for further analysis of their pleiotropic effects on multiple traits, such as GL, PL, and TGW, across multiple environments. First, the RIL population was divided into two genotypic groups, aa, and bb, based on the bin-marked genotypes within different QTL intervals, excluding the heterozygous type. The differences in corresponding traits between the two genotypic groups were then evaluated.

The results showed that RILs with favorable alleles exhibited higher average phenotypic values for all corresponding traits than RILs without favorable alleles. Some traits were detected with QTLs only in one or two environments, but differences between the aa and bb genotypic groups were observed across all three environments. Analysis of variance indicated that, except for the QTL *qTGW-2* controlled by Locus 1, which did not reach significance in the 2019 environment for TGW, the remaining traits showed significant or highly significant differences across all three environments (Table 5). This finding suggested that these three stable QTLs were reliable and could enhance the value of the corresponding traits.

To further confirm the combined effects of these three stable QTLs on six traits, including GL, GW, GLWR, TGW, AS, and PL, the RIL population was divided into eight combinations based on the combination of the three loci (Figure 3). Through analyzing the relationship between the phenotypic values and the number of favorable alleles in different combination types, the results were as follows. Except for GW and GLWR, all other phenotypic values increased with the number of favorable alleles, and some reached significant levels. It is worth noting that Hap1, which carries all three favorable genes, showed significant improvements in GL, TGW, AS, and PL compared to Hap2, Hap3, Hap4 (carrying two favorable genes), Hap5, Hap6, Hap7 (carrying one favorable gene), and Hap8 (carrying zero favorable genes). However, significant improvements were observed in the GL when analyzing the phenotypic values based on the presence of zero, one, or two favorable genes.

In contrast, the improvements in other traits did not reach significant levels. Our results indicated that the genetic effects of these three loci were stable and reliable, and the pyramiding of favorable alleles mainly exhibited an additive effect, effectively improving the phenotypic values of the correlated traits. However, if the goal is to significantly improve the phenotypic values of the correlated traits, only one or pyramiding of two favorable QTLs is unlikely to achieve the desired effect. It is necessary to pyramide all three favorable genes to achieve the ideal effect.

## 3. Discussion

Currently, most researchers employ traditional molecular markers, such as RFLP, SSR, and InDel, to construct genetic maps with an accuracy of approximately 1–10 Mb [38]. Due to limitations in the number and coverage density of polymorphic genetic markers, the mapping intervals are larger, and the precision is lower. Additionally, they cannot accurately detect reciprocal translocation breakpoints [39]. In contrast, the bin map is constructed based on sequencing technology, with a higher number and density of markers. It achieves a precision of up to 100 kb and provides accurate physical positions, enabling more precise QTL mapping with smaller intervals.

Furthermore, each bin contains multiple non-recombinant SNPs, effectively reducing the occurrence of missed reciprocal translocation breakpoints. In this study, we utilized whole-genome resequencing technology to construct a high-density genetic map based on bin markers, encompassing 2193 bin markers. The total map distance was 1542.27 cM, with an average physical distance of 201.29 kb between markers. QTL mapping was performed for rice GL, GW, GLWR, TGW, AS, and PL in three different environments. A total of 39 QTLs were identified, with mapping intervals ranging from 8.1 kb to 1781.6 kb and an average physical distance of 517.5 kb. Among them, 15 QTLs were detected in two or more environments. We obtained 14 stable genetic effect loci, including three loci that overlapped with previous reports and 11 loci that represented new QTLs consistently associated with two or more environments or traits. Compared to genetic maps constructed using traditional markers, the bin map exhibited significantly improved marker density, higher QTL mapping resolution, and enhanced positional accuracy. Additionally, previous studies have indicated that QTLs controlling related traits are not uniformly distributed across chromosomes; instead, many QTLs are closely linked or clustered in specific chromosomal regions [40]. In previous studies using traditional genetic maps for mapping, it is often challenging to precisely locate multiple adjacent loci within a segment where multiple peaks exceed the threshold LOD value, leading to the omission of some loci with smaller effects [41]. In our present study, we identified several QTLs located in close proximity on the chromosomes, such as *qGW-3-1* and *qGW-3-2* on chromosome 3, spanning 15.92–16.07 Mb and 16.15–16.62 Mb, respectively, as well as *qAS-4-2* and *qAS-4-3* on chromosome 4, spanning 24.10–24.11 Mb and 24.19–24.63 Mb, respectively. This finding illustrated that the bin map enabled more precise detection of QTLs, accurately pinpointing adjacent loci.

In recent years, numerous researchers have used different genetic populations and markers to identify and clone many QTLs related to grain size in rice. The reported pathways regulating rice grain size include the ubiquitin-proteasome system, G-protein signaling, mitogen-activated protein kinase (MAPK) signaling, plant hormone perception and homeostasis, transcriptional regulatory factors, and epigenetic modifications. It has been reported that *GGC2* affects GL and GW in rice through the G-protein pathway, thereby influencing the appearance quality and yield of rice [37]. In this study, *GGC2* was located in the Locus 13 region, which showed consistent expression and pleiotropic effects across multiple environments when assessing the traits of GL, TGW, and PL. Additionally, a QTL, *qGLWR-5*, associated with GLWR was mapped to chromosome 5, in the same position as the previously reported gene *OsPUP7*, which affects rice grain shape. *OsPUP7*, a cytokinin transport protein, plays a crucial role in various biological processes, such as cell division and differentiation, thereby regulating grain shape and size [36]. On chromosome 2, a QTL, *qAS-2-1*, controlling AS, was located at the same position as the previously reported gene *PGL2*, which influences rice grain size. *PGL2* encodes an atypical bHLH protein that regulates GL and GW in rice through interactions with the typical bHLH protein APG [35].

The utilization of the same mapping population across multiple environments for QTL mapping is highly significant for aggregation breeding. It enables the detection of stable QTLs with minimal environmental influence, which can be consistently identified. Additionally, the identification of pleiotropic QTLs is valuable in breeding, as multiple traits can be simultaneously selected using a single genomic region [42,43]. In this study, among the newly identified QTLs, Locus 14 was a stable QTL detected in multiple environments and simultaneously influenced multiple traits, such as GW, GLWR, and AS. It exhibited pleiotropy and had a high contribution rate, providing a foundation for gene cloning. Furthermore, three genetic pleiotropic loci were associated with GL, TGW, AS, and PL, namely Locus 1, Locus 3, and Locus 10. These loci not only stably expressed QTLs across multiple environments, but also demonstrated pleiotropy. All of these loci were important QTLs and provided opportunities for using marker-assisted selection (MAS) to improve rice grain yield and appearance traits.

Pyramide breeding is a viable approach for increasing yield. In previous studies, the aggregation analysis of multiple QTLs related to GL and GW, such as *GS3*, *GL2*/*OsGRF*/*GS24*, *GW8*, *GL3.1*/*ospplkl1*, and *GLW7*/*OsSPL13* has shown that introducing multiple favorable alleles can effectively increase GL and improve rice yield [6,10,13]. This study identified 14 stable genetic loci related to grain traits. Through the aggregation analysis of three of these loci, the results were consistent with our expected goals. The accumulation of multiple favorable alleles could effectively increase GL and improve rice yield (Figure 3), demonstrating that these loci could be applied in aggregation breeding in rice.

## 4. Materials and Methods

### 4.1. Plant Materials

In the present study, ZP37, a local variety collected by our research group, was characterized by strong and stout stems, numerous secondary branches, and strong disease resistance. R8605, on the other hand, was a hybrid rice restorer line developed by the Rice Research Institute of Guangxi Academy of Agricultural Sciences. It exhibited characteristics such as exceptionally long panicles, cold tolerance, and high-quality and high-yield traits. Using R8605 as the paternal parent and ZP37 as the maternal parent, an F_1_ hybrid was obtained, and subsequent generations were generated through multiple generations of self-pollination using the single seed descent method. This resulted in the construction of a set of RIL populations (F_8:9_ generation). Both the parental lines and RILs were planted in Nanning City, Guangxi, China (108°22′ E, 22°48′ N) for early-season cultivation in 2019 and late-season cultivation in 2020. Additionally, in 2022, they were also planted in Guiping City, Guangxi, China (114°07′ E, 22°32′ N). These planting locations were labeled as 2019, 2020, and 2022, respectively. The sowing and transplanting of the early-season planting took place on 1 March and 2 April each year, respectively, while those for the late-season planting occurred on 18 July and 4 August, respectively. Each RIL or parental line was arranged in a randomized complete block design, with six rows per line and six plants per row, spaced at 20 cm × 20 cm intervals. Single seedling transplanting was employed, and field management followed standard local agricultural practices.

### 4.2. Phenotype Investigation and Data Analysis

During the rice maturation stage, we collected three plants at random from each plot for harvesting. The harvested grains were naturally dried, and any empty grains were removed from the samples. To analyze the grain-related traits, we utilized the SC-G automatic seed phenotyping system (Hangzhou WSeen Detection Technology Co., Ltd., Hangzhou, China). The system was operated following the provided instructions, with precise settings and appropriate threshing thresholds to ensure accurate measurements. Six specific traits were analyzed in this study, including GW, GL, GLWR, area size (AS) of grain, perimeter length (PL) of grain, and TGW. The grains were then weighed using an electronic balance, and the average values from the three plants were recorded as the final phenotypic values for each trait.

The data were recorded, analyzed, and processed using WPS Office Excel. Histograms depicting the different traits of the RIL population across multiple environments were created. Statistical analysis was performed using DPS v9.01, and significant differences were detected using the LSD method. Correlation analysis was conducted using SPSS Statistics 26.

### 4.3. Genotype Identification and High-Density Genetic Map Construction of the RIL Population

During the tillering stage of rice, tender leaves of both the parental lines and the RIL population (F_8:9_ generation) were collected. The DNA extraction and sequencing analysis were entrusted to Biomarker Technologies Co., Ltd. (Beijing, China). The specific steps were as follows. First, DNA extraction was performed using the CTAB method. After sample validation, the DNA was randomly fragmented using the Covaris ultrasonic disruptor. The fragmented DNA was subjected to end repair, 3′ end adenylation, sequencing adapter ligation, purification, and PCR amplification to construct sequencing libraries. The Illumina sequencing platform was used to generate raw sequencing data, known as sequencing reads. The data underwent filtering steps to remove adapter contamination, reads with an N content exceeding 10%, and low-quality reads. This resulted in obtaining high-quality clean reads for subsequent analysis. The sequencing reads were aligned to the reference genome of the rice variety ‘Nipponbare’ (IRGSP-1.0) using the BWA software (http://bio-bwa.sourceforge.net/, accessed on 6 August 2023). Only uniquely aligned paired-end reads were retained. Subsequently, the GATK software (version 3.6-0-g89b7209) [44] was employed for SNP detection between the parental lines. Preprocessing steps were performed, such as duplicate marking using Picard and local realignment using GATK. SNP detection and filtering were conducted to obtain the final set of SNP loci. The genotypic data of the 208 RILs were statistically analyzed. The “sliding window” method [45] was employed to identify recombination breakpoints. Multiple consecutive SNP markers showing tight linkage without recombination in all samples were considered a single block (bin). The physical position of each bin was determined based on the starting point of the bin marker. Within each bin, markers with the same genotype as ZP37 were labeled “aa”, while markers with the same genotype as R8605 were labeled “bb”. Recombination rates were calculated using the maximum likelihood function based on the genotypes, and genetic distances were determined using the Kosambi mapping function. High-density genetic maps were constructed using the maximum likelihood estimation method in the HighMap software (Scmap V5). The best position for each marker was determined based on the optimal AIC (Akaike information criterion) value.

### 4.4. QTL Mapping

The high-density genetic map constructed based on whole-genome resequencing was used for trait mapping using the composite interval mapping (CIM) method. A significance threshold (PIN) of 0.001 was set, and the genome-wide scan was performed with a step size of 1 cM. The LOD threshold was determined through 1000 permutations using the PT (permutation test) method. Initially, a confidence level of 99% was set, and if no mapping interval was identified, it was reduced to 95% or 90%. If still no result was obtained, the PT test results were not considered, and the LOD threshold was manually lowered to 3.0. If no interval was identified at 3.0, it was further reduced to 2.5 or 2. The LOD value at the peak was considered the LOD value of the corresponding QTL. The effect of the QTL was estimated based on the bin marker at the peak position, and the additive effect and the contribution rate of each QTL to the trait were calculated. QTLs were named following the principles previously described [46]. A positive additive effect indicates that the enhancing allele originates from the ZP37 parental line, while a negative value indicates that it originates from the R8605 parental line.

### 4.5. Analysis of Genetically Stable Loci and Pyramiding Effects

The genotyping results of all bin markers within each mapping interval were analyzed. The genotypes of the 208 offspring in this interval were classified as “aa” and “bb” types. If different genotypes of bin markers were observed within the interval, it indicated the occurrence of recombination events, and such intervals were classified as heterozygous intervals. Stable QTL refers to the loci that can be repeatedly identified across multiple environments. The stable QTLs with higher contribution rates were selected. The distribution of enhancing alleles in the RIL population was analyzed. The RIL population was grouped based on the pyramiding of different favorable genes. The performance of traits in each group was evaluated to analyze the aggregation effect of different numbers of favorable genes.

## 5. Conclusions

In the present study, we constructed a high-density genetic map of a RIL population derived from the cross between R8605 and ZP37. This map consisted of 2193 bin markers with an average physical distance of 201.29 kb between markers. A total of 39 QTLs related to rice grain traits were identified in three different environments, with the mapping intervals ranging from 8.1 kb to 1781.6 kb and an average physical distance of 517.5 kb. By analyzing the genetic effects of the identified QTLs, 14 stable genetic loci (involving 30 QTLs) were obtained. Among them, four loci were novel loci exhibiting pleiotropic effects, controlling at least three traits and detected in multiple environments. Furthermore, pyramiding effects analysis of three loci showed that as the number of QTL increased, the phenotypic values of grain traits could be effectively improved. Collectively, our newly identified QTLs could be used in future breeding programs to enhance rice yield.

## Figures and Tables

**Figure 1 plants-12-02911-f001:**
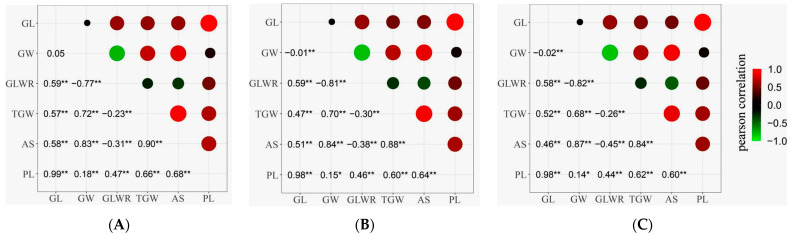
Correlation analysis of six grain traits in RILs. (**A**) 2019; (**B**) 2020; (**C**) 2022. In the upper panel, the size of the circle and depth of shading indicate the magnitude of correlations. Green are negative correlations, and red are positive correlations. The lower diagonal shows the correlation coefficients, *: *p* < 0.05, **: *p* < 0.01. The diagonal represents the frequency distribution of the six traits.

**Figure 2 plants-12-02911-f002:**
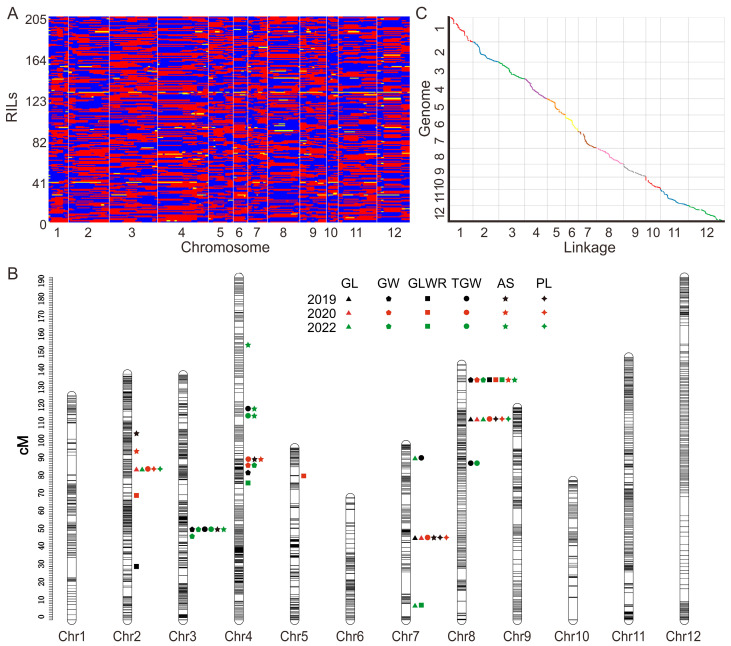
Construction of the high-density genetic map of the RIL population. (**A**): The high-density Bin marker genotype of the RIL populations. red: ZP37 genotype; blue: R8605 genotype; yellow: heterozygote. (**B**): Distribution of genetic markers and QTLs on chromosomes; (**C**): The correlation between the linkage marker on the genetic map and the physical map, respectively.

**Figure 3 plants-12-02911-f003:**
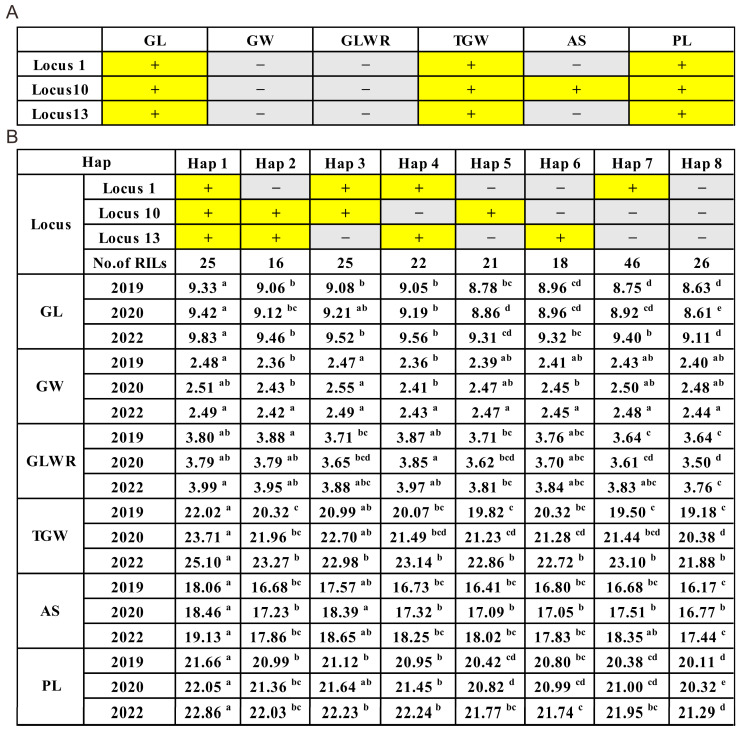
Pyramiding effects analysis of three stable QTLs. (**A**) Summary of the traits associated with the three QTLs. A yellow background indicates that the locus is associated with the trait, while a gray background indicates that the locus is not associated with the trait. (**B**) Pyramiding effects for different numbers of favorable alleles of the QTL. Letters from a to e indicate significantly different values according to statistical analysis (a = 0.05).

**Table 1 plants-12-02911-t001:** Phenotypes of ZP37, R8605, and the ZP37 × R8605 RIL population across cropping seasons.

Trait ^a^	Environment ^b^	Parents ^c^		RIL Population				
		ZP37	R8605	Mean	Range	Skewness	Kurtosis	CV ^d^ (%)
GL (mm)	2019	8.56 ± 0.21	9.27 ± 0.10 **	8.91	7.96–10.23	0.42	−0.06	4.78
	2020	8.68 ± 0.22	9.49 ± 0.12 **	9.02	7.90–10.29	0.08	0.25	4.65
	2022	8.80 ± 0.09	9.54 ± 0.22 **	9.43	8.48–10.70	0.30	0.13	4.37
GW (mm)	2019	2.40 ± 0.06	2.55 ± 0.02 *	2.42	2.08–2.75	−0.02	−0.65	6.06
	2020	2.45 ± 0.03	2.57 ± 0.08	2.48	2.12–2.97	0.00	−0.21	6.29
	2022	2.52 ± 0.04	2.61 ± 0.09	2.46	2.09–2.98	0.21	−0.02	6.30
GLWR	2019	3.57 ± 0.02	3.63 ± 0.03 *	3.72	3.15–4.55	0.33	−0.35	7.60
	2020	3.55 ± 0.06	3.70 ± 0.07 *	3.67	2.99–4.68	0.39	0.00	8.02
	2022	3.49 ± 0.05	3.66 ± 0.05 *	3.87	3.04–4.81	0.45	0.15	7.86
TGW(g)	2019	17.36 ± 0.11	21.95 ± 0.34 **	20.20	15.19–28.69	0.32	0.13	11.08
	2020	18.22 ± 0.17	23.40 ± 0.21 **	21.76	16.23–28.35	0.07	−0.03	10.17
	2022	19.70 ± 0.35	24.47 ± 0.14 **	23.13	15.16–29.95	−0.07	0.32	10.37
AS (mm^2^)	2019	16.25 ± 0.06	17.96 ± 0.23 **	16.89	13.77–20.70	0.05	−0.49	8.18
	2020	16.53 ± 0.11	18.64 ± 0.30 **	17.53	13.95–21.07	−0.10	−0.17	8.14
	2022	16.95 ± 0.19	19.39 ± 0.25 **	18.24	14.76–22.81	0.15	−0.21	8.23
PL (mm)	2019	19.89 ± 0.10	21.36 ± 0.22 **	20.73	18.68–23.63	0.43	−0.05	4.48
	2020	20.08 ± 0.19	21.57 ± 0.18 **	17.53	18.97–23.57	0.04	−0.02	4.16
	2022	21.19 ± 0.013	22.42 ± 0.37 **	22.00	19.93–24.53	0.17	−0.06	4.00

^a^ Trait: GL, grain length; GW, grain width; GLWR, length-to-width ratio of grain; TGW, 1000-grain-weight; AS, area size of grain; PL, perimeter length of grain. ^b^ Environment: 2019 is the early season in 2019, Nanning; 2020 is the late season in 2020, Nanning; 2022 is the late season in 2022, Guiping. ^c^ Parent refers to the mean ± standard deviation (SD) of the parents, *: *p* < 0.05, **: *p* < 0.01. ^d^ CV (%), coefficient of variation.

**Table 2 plants-12-02911-t002:** Distribution of genetic markers across the 12 chromosomes in rice.

Chromosome	Number of Bin Markers	Length (cM)	Average Genetic Distance between Markers (cM)	Average Physical Distance between Markers (kb)
1	122	126.24	1.03	354.65
2	248	138.35	0.56	144.88
3	293	137.93	0.47	124.27
4	308	192.93	0.63	115.14
5	151	96.82	0.64	198.39
6	88	68.68	0.78	355.05
7	120	98.58	0.82	247.34
8	196	143.89	0.73	145.10
9	162	119.46	0.74	142.04
10	71	78.38	1.10	326.85
11	235	147.96	0.63	123.47
12	199	193.02	0.97	138.34
Total	2193	1542.24	0.76	201.29

**Table 3 plants-12-02911-t003:** QTLs for grain size-related traits of recombinant inbred lines in different environments.

Trait	QTL	Chr.	Marker Interval	Physical Interval (bp)	QTL Interval Size (kb)	Environment	LOD	PVE (%)	ADD
GL	*qGL-2*	2	Block1305–Block1318	29,677,627–30,291,539	613.9	2020	5.56	6.50	0.16
		2	Block1305–Block1318	29,677,627–30,291,539	613.9	2022	4.81	5.39	0.14
	*qGL-7-1*	7	Block5298–Block5303	2,305,563–2,667,445	361.9	2022	3.43	2.16	−0.09
	*qGL-7-2*	7	Block5433–Block5439	20,606,977–21,226,418	619.4	2019	4.68	5.72	−0.15
		7	Block5433–Block5439	20,606,977–21,226,418	619.4	2020	5.53	4.98	−0.14
	*qGL-7-3*	7	Block5588–Block5589	28,557,384–28,724,630	167.2	2022	2.85	1.74	−0.08
	*qGL-8*	8	Block6010–Block6012	23,379,921–23,447,755	67.8	2019	4.85	6.96	−0.17
		8	Block5989–Block6010	22,065,235–23,387,924	1322.7	2020	5.56	6.12	−0.15
		8	Block5989–Block6010	22,065,235–23,387,924	1322.7	2022	4.81	3.46	−0.11
GW	*qGW-3-1*	3	Block1966–Block1967	15,915,064–16,067,455	152.4	2022	3.00	3.29	0.04
	*qGW-3-2*	3	Block1968–Block1974	16,148,731–16,617,377	468.6	2019	4.81	4.93	0.05
		3	Block1970–Block1974	16,148,731–16,617,377	468.6	2022	3.00	3.90	0.05
	*qGW-4-1*	4	Block3702–Block3701	19,871,103–19,942,280	71.2	2019	4.81	3.23	−0.04
	*qGW-4-2*	4	Block3808–Block3815	20,702,635–20,878,645	176.0	2020	4.89	5.89	−0.06
		4	Block3808–Block3815	20,702,635–20,878,645	176.0	2022	3.00	3.29	−0.04
	*qGW-8*	8	Block6167–Block6169	27,555,403–27,911,017	355.6	2019	8.91	9.21	0.07
		8	Block6167–Block6169	27,555,403–27,911,017	355.6	2020	15.71	14.10	0.09
		8	Block6167–Block6169	27,555,403–27,911,017	355.6	2022	11.57	10.73	0.07
GLWR	*qGLWR-2-1*	2	Block821	9,286,980–10,372,621	1085.6	2019	3.88	2.91	0.07
	*qGLWR-2-2*	2	Block1221–Block1270	26,355,325–27,386,770	1031.4	2020	3.68	2.03	0.06
	*qGLWR-4*	4	Block3684–Block3688	19,157,777–19,344,941	187.2	2022	3.00	2.01	0.06
	*qGLWR-5*	5	Block4850–Block4851	27,634,637–27,858,389	223.8	2020	3.68	3.53	0.08
	*qGLWR-7*	7	Block5298–Block5299	2,305,563–2,328,233	22.7	2022	3.00	1.45	−0.05
	*qGLWR-8*	8	Block6167–Block6169	27,555,403–27,911,017	355.6	2019	16.30	14.20	−0.16
		8	Block6167–Block6169	27,555,403–27,911,017	355.6	2020	15.51	13.83	−0.16
		8	Block6167–Block6169	27,555,403–27,911,017	355.6	2022	16.12	12.46	−0.16
TGW	*qTGW-2*	2	Block1305–Block1318	29,677,627–30,291,539	613.9	2020	4.85	4.97	0.41
	*qTGW-3*	3	Block1970–Block1974	16,148,731–16,617,377	468.6	2019	3.29	2.66	0.05
		3	Block1970–Block1974	16,148,731–16,617,377	468.6	2022	4.70	3.67	0.67
	*qTGW-4-1*	4	Block3830–Block3834	20,994,282–21,294,165	299.9	2020	3.97	3.92	−0.64
	*qTGW-4-2*	4	Block3948–Block3949	24,101,830–24,109,967	8.1	2022	4.89	3.76	−0.68
	*qTGW-4-3*	4	Block3955–Block3957	24,194,456–24,626,257	431.8	2019	4.54	2.93	−0.56
	*qTGW-7-1*	7	Block5435–Block5490	20,956,479–22,680,384	1723.9	2020	3.97	4.23	−0.67
	*qTGW-7-2*	7	Block5588–Block5589	28,557,384–28,724,630	167.2	2019	3.00	2.09	−0.48
	*qTGW-8-1*	8	Block5806–Block5808	19,645,419–19,698,589	53.2	2019	4.79	3.76	−0.64
		8	Block5807–Block5808	19,672,283–19,698,589	263.1	2022	4.89	5.75	−0.85
	*qTGW-8-2*	8	Block5989–Block6012	22,065,235–23,447,755	1382.5	2020	4.85	4.60	−0.70
AS	*qAS-2-1*	2	Block1352–Block1354	31,122,113–31,447,924	325.8	2020	4.88	5.34	0.48
	*qAS-2-2*	2	Block1376–Block1377	32,614,759–32,704,042	89.3	2019	3.60	4.07	0.41
	*qAS-3*	3	Block1970–Block1974	16,148,731–16,617,377	468.6	2019	3.78	3.38	0.37
		3	Block1970–Block1974	16,148,731–16,617,377	468.6	2022	2.87	4.25	0.45
	*qAS-4-1*	4	Block3830–Block3834	20,994,282–21,294,165	299.9	2019	3.39	3.42	−0.38
		4	Block3830–Block3834	20,994,282–21,294,165	299.9	2020	4.88	4.77	−0.46
	*qAS-4-2*	4	Block3948–Block3949	24,101,830–24,109,967	8.1	2022	3.00	3.55	−0.42
	*qAS-4-3*	4	Block3955–Block3967	24,194,456–25,976,047	1781.6	2022	3.00	3.31	−0.40
	*qAS-4-4*	4	Block4088–Block4087	30,322,633–30,361,362	38.7	2022	3.00	2.45	−0.35
	*qAS-7*	7	Block5435–Block5463	20,956,479–21,435,416	478.9	2019	3.35	3.51	−0.38
	*qAS-8*	8	Block6167–Block6169	27,555,403–27,911,017	355.6	2020	9.17	8.08	0.60
		8	Block6167–Block6169	27,555,403–27,911,017	355.6	2022	5.70	6.30	0.55
PL	*qPL-2*	2	Block1305–Block1318	29,677,627–30,291,539	613.9	2020	5.03	6.91	0.34
		2	Block1305–Block1318	29,677,627–30,291,539	613.9	2022	4.93	5.79	0.31
	*qPL-7*	7	Block5433–Block5439	20,606,977–21,226,418	619.4	2019	4.80	5.85	−0.33
		7	Block5433–Block5435	20,606,977–20,956,479	349.5	2020	5.03	5.83	−0.31
	*qPL-8*	8	Block5989–Block6010	22,065,235–23,387,924	1322.7	2019	4.93	6.06	−0.34
		8	Block5989–Block6012	22,065,235–23,447,755	1382.5	2020	5.03	4.44	−0.27
		8	Block5989–Block6010	22,065,235–23,387,924	1322.7	2022	3.41	3.12	−0.23

LOD, logarithm of odds; PVE (%), phenotypic variation explained (%); ADD, additive effect; a positive value indicates an increasing effect from parent ZP37.

**Table 4 plants-12-02911-t004:** Major QTL clusters associated with grain shape traits detected in this study.

S. No.	Locus	QTL	Marker Interval	Physical Interval (bp)	PVE (%)	Overlapped QTL Reported
1	Locus 1	*qGL-2* (2020, 2022); *qTGW-2* (2020); *qPL-2* (2020, 2022)	Block1305–Block1318	29,677,627–30,291,539	4.97–6.91	
2	Locus 2	*qAS-2-1* (2020)	Block1352–Block1354	31,122,113–31,447,924	5.34	*PGL2*
3	Locus 3	*qGW-3-2* (2019, 2022); *qTGW-3* (2019, 2022); *qAS-3* (2019, 2022)	Block1968–Block1974	16,148,731–16,617,377	2.66–4.93	
4	Locus 4	*qGW-4-2* (2020, 2022)	Block3808–Block3815	20,702,635–20,878,645	3.29–5.89	
5	Locus 5	*qTGW-4-1* (2020); *qAS-4-1* (2019, 2020)	Block3830–Block3834	20,994,282–21,294,165	3.42–4.77	
6	Locus 6	*qTGW-4-2* (2022); *qAS-4-2* (2022)	Block3948–Block3949	24,101,830–24,109,967	3.55–3.76	
7	Locus 7	*qTGW-4-3* (2019); *qAS-4-3* (2022)	Block3955–Block3967	24,194,456–25,976,047	2.93–3.31	
8	Locus 8	*qGLWR-5* (2020)	Block4850–Block4851	27,634,637–27,858,389	3.53	*OsPUP7*
9	Locus 9	*qGL-7-1* (2022); *qGLWR-7* (2022)	Block5298–Block5303	2,305,563–2,667,445	1.45–2.16	
10	Locus 10	*qGL-7-2* (2019, 2020); *qTGW-7-1* (2020); *qAS-7* (2019); *qPL-7* (2019, 2020)	Block5433–Block5490	20,606,977–22,680,384	3.51–5.85	
11	Locus 11	*qGL-7-3* (2022); *qTGW-7-2* (2019)	Block5588–Block5589	28,557,384–28,724,630	1.74–2.09	
12	Locus 12	*qTGW-8-1* (2019, 2022)	Block5807–Block5808	19,672,283–19,698,589	3.76–5.75	
13	Locus 13	*qGL-8* (2019, 2020, 2022); *qTGW-8-2* (2020); *qPL-8* (2019, 2020, 2022)	Block5989–Block6012	22,065,235–23,447,755	3.42–6.96	*GGC2*
14	Locus 14	*qGW-8* (2019, 2020, 2022); *qGLWR-8* (2019, 2020, 2022); *qAS-8* (2020, 2022)	Block6167–Block6169	27,555,403–27,911,017	6.30–14.20	

**Table 5 plants-12-02911-t005:** Summary of the phenotypic effects of three stable QTL.

Trait	QTL	Environment	Number of RILs of Marker Type aa	Number of RILs of Marker Type bb	Donors of Positive Allele	Phenotypic Value		Difference
						Marker Type aa	Marker Type bb	
Locus 1								
GL (mm)	-	2019	82	120	bb	8.82 ± 0.37	9.00 ± 0.45	0.17 **
	*qGL-2*	2020				8.85 ± 0.37	9.13 ± 0.42	0.28 **
	*qGL-2*	2022				9.27 ± 0.38	9.54 ± 0.40	0.28 **
TGW(g)	-	2019				19.82 ± 2.09	20.42 ± 2.34	0.60
	*qTGW-2*	2020				21.11 ± 2.18	22.18 ± 2.18	1.07 **
	-	2022				22.57 ± 2.53	23.51 ± 2.17	0.94 **
PL (mm)	-	2019				20.51 ± 0.79	20.90 ± 0.99	0.39 **
	*qPL-2*	2020				20.81 ± 0.80	21.43 ± 0.86	0.62 **
	*qPL-2*	2022				21.65 ± 0.84	22.26 ± 0.82	0.61 **
Locus 10								
GL (mm)	*qGL-7-2*	2019	88	118	aa	9.07 ± 0.42	8.80 ± 0.39	0.27 **
	*qGL-7-2*	2020				9.17 ± 0.42	8.91 ± 0.38	0.26 **
	-	2022				9.55 ± 0.42	9.34 ± 0.39	0.21 **
TGW(g)	-	2019				20.87 ± 2.14	19.69 ± 2.18	1.18 **
	*qTGW-7-1*	2020				22.49 ± 2.19	21.2 ± 2.09	1.29 **
	-	2022				23.61 ± 2.50	22.72 ± 2.24	0.90 **
AS (mm^2^)	*qAS-7*	2019				17.27 ± 1.26	16.60 ± 1.41	0.66 **
	-	2020				17.89 ± 1.39	17.25 ± 1.40	0.64 **
	-	2022				18.51 ± 1.62	18.01 ± 1.38	0.49 *
PL (mm)	*qPL-7*	2019				21.08 ± 0.90	20.47 ± 0.87	0.60 **
	*qPL-7*	2020				21.51 ± 0.87	20.93 ± 0.80	0.58 **
	-	2022				22.26 ± 0.89	21.80 ± 0.83	0.46 **
Locus 13								
GL (mm)	*qGL-8*	2019	83	122	aa	9.11 ± 0.43	8.79 ± 0.37	0.32 **
	*qGL-8*	2020				9.19 ± 0.38	8.90 ± 0.41	0.30 **
	*qGL-8*	2022				9.56 ± 0.42	9.35 ± 0.39	0.21 **
TGW(g)	-	2019				20.75 ± 2.46	19.83 ± 2.02	0.92 **
	*qTGW-8-2*	2020				22.23 ± 2.44	21.45 ± 2.01	0.77 *
	-	2022				23.58 ± 2.60	22.86 ± 2.22	0.73 *
PL (mm)	*qPL-8*	2019				21.12 ± 0.95	20.47 ± 0.82	0.65 **
	*qPL-8*	2020				21.51 ± 0.79	20.94 ± 0.87	0.57 **
	*qPL-8*	2022				22.25 ± 0.88	21.84 ± 0.84	0.41 **

‘‘-” indicates that no QTL was detected during the season. ‘‘aa” indicates the genotype of ZP37. ‘‘bb” indicates the genotype of R8605. The difference of phenotypes of favorable alleles minus that of an unfavorable allele; *: *p* < 0.05, **: *p* < 0.01.

## Data Availability

The original contributions presented in the study are publicly available. The data that support the findings of this study have been deposited into CNGB Sequence Archive (CNSA) [47] of China National GeneBank DataBase (CNGBdb) [48] with accession number CNP0004463.

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
