# Peer review of "Construction of High-Density Genetic Map and QTL Mapping for Grain Shape in the Rice RIL Population"

_plants, 2023, doi:10.3390/plants12162911_

Round 1
Reviewer 1 Report
The paper reveals that a high-density genetic map for 208 RILs between R8605 X ZP37 consisting of 2,193 bin (526,957 SNP) markers with a total genetic distance of 1,542.27 cM (markers was 0.76 cM and 201.29 kb); A total of 39 QTLs intervals ranging from 8.1 kb to 1781.6 kb, especially four novel Locus(1, 3, 10, 14) exhibited pleiotropic effects on at least three traits, Locus 14 influenced grain width, length-to-width ratio, and grain cross-sectional area. These results provided a theoretical basis and genetic resources for the cloning, functional analysis, and molecular breeding of genes related to rice grain shape. The analysis process is comprehensive, good organized, advanced research technology, large amount of information and so on. Minor revision can be published in Plants. However, there are some major issues need to be improved:
1. Abstract: The abstract should be modified to enhance the readability;
2. Introduction: Some latest references should add about QTL mapping for grain shape based on SNP markers.
3. Results: It is better to occupy 2 rows adjust 1 row in the table 3 and Table 5.
4. DiscussioMaterials and Methodsn:There is a typo in line 345;The correlation mechanism of pleiotropic effects and grain shape?
5. Materials and Methods:The difference in the shape of the grains of the parents is not particularly large, how to avoid limitations in the method?
6. References: The full text was written in a Plants format template whenever possible.
Good
Author Response
Dear reviewer,
Thank you for your positive comments and valuable suggestions to improve the quality of our manuscript. The main corrections in the paper and the responds to your comments were as flowing.
Response to Reviewer 1 Comments
Point 1: Abstract: The abstract should be modified to enhance the readability;
Reponse 1: We have revised the abstract based on the reviewers' comments and suggestions.
Point 2: Introduction: Some latest references should add about QTL mapping for grain shape based on SNP markers.
Reponse 2: As suggested by the reviewer, we added three latest references ( References 23-25 ) about QTL mapping for grain shape based on SNP markers to the introduction part in the revised manuscript.
Point 3: Results: It is better to occupy 2 rows adjust 1 row in the table 3 and Table 5.
Reponse 3: As suggested by the reviewer, we adjusted 2 rows to 1 row by placing Tables 3 and 5 horizontally.
Point 4: Discussion: There is a typo in line 345; The correlation mechanism of pleiotropic effects and grain shape?
Reponse 4A: We feel sorry for our carelessness. The “high4er” has been corrected with “higher” in line 345 of the revised manuscript.
Reponse 4B: In this paper, QTL analysis was carried out for six traits (rice grain length, grain width, length-to-width ratio, thousand-grain weight, grain cross-sectional area, and grain perimeter) related to grain shape, of which Locus 1, Locus 3, Locus 10, and Locus 14 were novel loci exhibiting pleiotropic effects on at least three traits and were detected in multiple environments, and the locus with pleiotropic effects can increase seed volume and improve seed filling can increase seed yield in rice. Which are important factors in the composition of rice yield. In previous studies, some QTL controlling grain shape exhibit pleiotropic effects, simultaneously affecting multiple traits related to grain shape. For example, the GL7/GW7 gene located on chromosome 7 influences both GL and GW. Upregulation of GL7 leads to decreased transverse cell division and increased longitudinal cell division, resulting in thinner and longer grains with improved appearance (References 18,19). Another example is the GL2/GS2 gene located on chromosome 2, which is a major QTL that simultaneously influences GL, GW, and grain weight in rice (References5, 20). Additionally, genes like GW6a (References21) and GS9 (References22) also exhibit pleiotropic effects and play crucial roles in grain shape.
Reponse 5: Materials and Methods:The difference in the shape of the grains of the parents is not particularly large, how to avoid limitations in the method?
Reponse 5: Although the shape of the grains of the parents is not particularly large, a total of 39 QTLs were detected in our paper, of which 15 were repeatedly detected in more than two different environments, suggesting that the method of locating QTLs for grain shape in our genetic population by constructing a high-density genetic map is effective. In order to ensure that this method is not limited to the detection of minor QTLs, we believe that the genetic population should be as large as possible (208 RILs in the manuscript were suitable), and the random error in phenotyping should be minimized. Of course, for traits with high environmental impact and large genotypic and environmental interactions, phenotyping should also be performed in multiple locations/multiple years .
Point 6: References: The full text was written in a Plants format template whenever possible.
Reponse 6: We have revised and corrected all references according to the format of the "Plants" journal.

Reviewer 2 Report
I enjoyed reading this excellent manuscript. My comments are very few and minor.
Line 345 has a typo, "high4er"
The paragraph starting with "In recent years" at line 372 is especially interesting and greatly substantiates the significance of the results.
Author Response
Dear reviewer,
We are very grateful to you for your tremendous time and effort in reviewing our paper, as well as for your positive comments of our manuscript.
Response to Reviewer 2 Comments
Point 1: Line 345 has a typo, "high4er"
Response 1:We feel sorry for our carelessness. The “high4er” has been corrected with “higher” in line 345 of the revised manuscript.
Point 2:The paragraph starting with "In recent years" at line 372 is especially interesting and greatly substantiates the significance of the results.
Response 2:In recent years, it is true that there has been an increasing number of researchers using high-density genetic maps for QTL localization for grain shape. The examples cited in our paper are all successful cases (References 23-25 ).
